# Parental Practices and Environmental Differences among Infants Living in Upper-Middle and High-Income Countries: A Cross-Sectional Study

**DOI:** 10.3390/ijerph191710833

**Published:** 2022-08-31

**Authors:** Carolina Fioroni Ribeiro da Silva, Valentina Menici, Eloisa Tudella, Elena Beani, Veronica Barzacchi, Roberta Battini, Alessandro Orsini, Giovanni Cioni, Giuseppina Sgandurra

**Affiliations:** 1Postgraduate Program in Physiotherapy (PPG-Ft), Department of Physiotherapy, Nucleus of Studies in Neuropediatrics and Motricity (NENEM), Federal University of São Carlos (UFSCar), São Carlos 13565-905, Brazil; 2Department of Developmental Neuroscience, IRCCS Fondazione Stella Maris, Viale del Tirreno 331, 56128 Pisa, Italy; 3Ph.D. Programme in Clinical and Translational Sciences, University of Pisa, Via Roma 67, 56126 Pisa, Italy; 4Tuscan Ph.D. Programme of Neuroscience, University of Florence, 50121 Florence, Italy; 5Department of Clinical and Experimental Medicine, University of Pisa, Via Roma 67, 56126 Pisa, Italy; 6Pediatric Neurology, Pediatric Department, Santa Chiara University Hospital, Azienda Ospedaliero Universitaria Pisana, 56126 Pisa, Italy

**Keywords:** risk factors, environment and public health, stress, parental practices

## Abstract

Parental practices and environmental factors can impact a child’s development and, consequently, functionality. The objective is to assess the parental practices and environmental differences in healthy and at-risk infants at 3–6 months of age living in upper-middle (Brazil) and high-income (Italy) countries. A total group of 115 infants was identified and classified into four groups: healthy Italian infants (H_IT); Italian infants exposed to biological risk factors (R_IT); healthy Brazilian infants (H_BR); and Brazilian infants exposed to environmental risk factors (L_BR). The dependent variables were parental practices and environmental factors, which were assessed through a semi-structured interview and the “variety of stimulation dimension” from the Affordances in the Home Environment for Motor Development—Infant Scale (AHEMD-IS) questionnaire. Descriptive analyses, a multivariate analysis of variance (MANOVA), and correlation tests were applied. Regarding the environment and parental practices, the mother’s age, maternal and paternal education, civil status, and variety of stimulation showed significant differences among the infants living in Brazil or in Italy. There were strong dissimilarities in parental practices and environmental factors among infants living in low/upper-middle and high-income countries. Since the home environment is the main stimulus for infant growth and development, our results are meaningful for providing knowledge about these two different cultures.

## 1. Introduction

An infant’s nervous system is not a simple collection of reflexes, but a complex system, characterized by high brain plasticity, which can manage its motor behavior according to its adaptive needs and response to its interaction with the environment [1,2]. Infants are more susceptible to receive, interpret, and respond to sensory inputs and environmental demands during the first 1000 days of life, leading to the development of body structures, functions, and new skills [3]. This period is named the sensitive period because, at this time, a high level of neuroplasticity is present and, consequently, the infants are in an intensive process of neurogenesis, migration, synaptogenesis, axonal and dendritic growth, [4] and therefore able to receive, interpret, and respond to extrinsic stimuli [5,6]. In this way, parental practices, maternal and paternal education, socioeconomic status (SES), and the variety of stimulation offered in the home environment, i.e., extrinsic factors, can impact the child’s development, affecting motor and cognitive skills, learning, behavior, and, consequently, functionality [7,8].

When it comes to extrinsic factors, low SES is an alarming environmental factor worldwide. In Italy, a high-income country, 1.9 million people have been reported as living in poverty, representing about 7.5% of the total population [9]. Regarding the number of children, approximately 1.337 million children are harmed by poverty [10]. In upper-middle-income countries (UMIC), the picture is more complex, since 1.3 billion people live in multidimensional poverty [11], lacking healthcare, education, housing, political participation, gender equality, and economic and basic elements [12]. Furthermore, UMICs have about 644 million multidimensional poor children, higher than in high-income countries [11]. In Brazil, until the beginning of 2020, 40% of children and adolescents lived in poverty, and about 12% in extreme poverty. According to the United Nations International Children’s Emergency Fund (UNICEF), and the United Nations (UN), extreme poverty proportionally doubles the impact on children and adolescents, compared to adults [13].

Low SES is associated with home chaos and the impaired socio-emotional, cognitive, and behavioral development of infants [14]. Home chaos is a disorganized and noisy environment that lacks a family routine, planning, and structure to perform daily activities [15]. In addition, it was observed that children from families in poverty can present alterations structural brain patterns, such as a reduction of the gray matter in the frontal, temporal and parietal lobes, and in the hippocampus [16]. This finding occurs, therefore, due to the events related to a low socioeconomic status, such as continuous noise, a low educational level, and the caregivers’ stress problems; in this way, these factors can interfere with attention and emotional levels [17].

The researchers show that the improvement of family stress and, consequently, parental care practices can produce an environment that promotes healthy development, even though infants are exposed to socioeconomic risk factors [18,19]. However, different cultural practices can interfere with motor development [20,21], and the countries present different ways to stimulate motor development and interact with infants (Karasik et al. 2015 [22]). It was observed, in a multicentric study, that infants who were born in Ghana and infants who were born in Norway presented the emergence of independent sitting at different ages, being at 5 and 7 months of age, respectively. The explanation for this fact may be that, in Ghana, mothers spontaneously provided a variety of stimuli in the sitting position, which helped the infant to acquire this skill. Conversely, Norwegian mothers did not encourage their children to perform the sitting posture as soon as possible [23].

The environment at home is crucial, and essential for the first months of infancy, due to the presence of stimuli for learning, responsive care, and organization in the home environment. Positive correlations were observed between parental practices, an adequate environment, and child development; infants and children aged 4 to 26 months succeeded in the domains of language, and in gross motor and socio-emotional development [24].

A variety of stimulation through parenting practices such as infant physical activity (e.g., tummy time) was longitudinally associated with advanced child development [25]. However, a study carried out in the United Kingdom, a high-income country, showed that just over 30% of families place their infants in the tummy time position [26]. Therefore, an Australian research group is studying strategies to make families aware and increase their adherence to adequate parental practices [27].

However, in UMICs such as Brazil, the mothers may have difficulties attending prenatal visits and maintaining a healthy pregnancy, probably because most are teenagers facing an unplanned pregnancy or lacking proper feeding, due to reduced economic resources [28]. Conversely, it is estimated that, in Italy, 94.3% of the mothers attend prenatal visits and receive health orientation from the third month of pregnancy [29]. In addition, it is known that biological risk factors, such as gestational age, increase the chances of the infant being at risk for neuromotor disorders. Approximately every year, 15 million babies are born preterm (before 37 completed weeks of gestation), and this number is rising. Preterm infants, even in high-income countries, are exposed to an increased risk of adverse neonatal outcomes and long-term neurodevelopmental and behavioral sequelae, lower cognitive functioning, and ongoing respiratory and other morbidities [30]. However, a suitable home environment shows benefits in cognitive ability, even for very preterm infants [31].

The environment plays an important role in neuromotor development, e.g., the earlier you optimize the home environment, the better the results in neuromotor development [32,33]. An assessment in order to create tools to improve the parental practices and environment is essential for providing the facilities that children need to fully develop during the first months of life, the sensitive period [34]. During this period, infants are in a constant process of neuroplasticity, that is, intense neurogenesis, migrations, neuronal, synaptogenesis, myelination, and axonal and dendritic growth, that is, able to receive, interpret and respond to extrinsic stimuli [6].

Thus, seeking more knowledge regarding parental practices and environmental factors in different cultures, SES, and populations, this study aimed to assess the parental practices and environmental differences in healthy and at-risk infants at 3–6 months of age living in upper-middle (Brazil) and high-income (Italy) countries.

Based on the research reported, it was hypothesized that Brazilian infants exposed to a lower socioeconomic level would show significantly different environment and parenting practices when compared to Italian infants.

## 2. Materials and Methods

### 2.1. Study Design

This is a retrospective, cross-sectional study, which follows the STROBE statement guidelines for observational studies [35].

### 2.2. Participants

A multicenter, retrospective study was designed involving the Department of Physiotherapy, Nucleus of Studies in Neuropediatrics and Motricity (NENEM) in Brazil, and the Department of Developmental Neuroscience of Stella Maris Foundation Italian Research and Clinical Scientific Institute (IRCCS) (Pisa) in Italy.

For the present retrospective study, infants and their caregivers were selected from medical records of basic health units, patient database collection, and from the maternity hospital of a medium-sized city. All the infants’ caregivers who attended the inclusion criteria were invited by phone or message to participate in the study.

The inclusion criteria were: (i) the gestational age of infants between 28 + 0 weeks and 42 + 0 weeks, and (ii) a clinical evaluation available at the infants’ corrected age between 3 and 6 months. The exclusion criteria defined were: (i) the presence of a brain injury, (ii) infants born small for their gestational age, (iii) a history of seizures, and (iv) a known genetic or metabolic disease.

The caregivers of these infants who agreed to answer the semi-structured interview were included in the study, and all of them agreed to participate. The study does not present missing data and dropouts. The infants were divided into four groups according to their characteristics, as reported by the caregiver or present in their medical records:-Healthy Italian infants (H_IT): Italian infants who were not exposed to environmental risk (i.e., low SES), and were not diagnosed with any health conditions when they were evaluated. The infants were not exposed to biological risks (i.e., without prenatal (small for their gestational age, suggesting intrauterine growth restriction), and perinatal (anoxia and hypoxia) or postnatal (neurological, hearing, visual, or sensory deficits, genetic syndromes, and musculoskeletal and cardiac abnormalities) impairments.-Italian infants exposed to biological risk factors (R_IT): Italian infants prematurely born (≤37 weeks) [8].-Healthy Brazilian infants (H_BR): composed of Brazilian infants who were not exposed to environmental risk, as written for the H_IT.-Brazilian infants exposed to environmental risk factors (L_BR): Brazilian infants exposed to low SES.

### 2.3. Procedures and Dependent Variables

The caregivers of infants were recruited by voluntary participation. The semi-structured interview was carried out face-to-face or remotely, according to the caregivers’ preference. It is divided into two parts:Environmental factors: information regarding the infant’s and caregiver’s data, selected by the authors of this study. The dependent variables were: maternal age, number of children at home, number of adults at home, SES, maternal and paternal education, and civil status. The variables have been made dichotomous and qualitative (yes/no), such as the SES and civil status. Specifically, the SES for Brazilian families was determined by considering the income-to-poverty ratio (PIR) [36] and, for Italian families, according to the calculation of the poverty threshold as defined by the National Institute of Statistics [37]. In order to equalize the groups according to the countries’ income, both tools take into account the poverty level specific to the area of residence. The level of maternal and paternal education was classified according to the International Standard Classification of Education (ISCED) [38] and categorized as low (ISCED levels < 3), intermediate (ISCED levels 3–4), or high (tertiary education, ISCED levels 5–8).Parental practices: the area of “variety of stimulation dimension”, from the Affordances in the Home Environment for Motor Development—Infant Scale (AHEMD-IS) questionnaire [39]. It includes questions about the amount of time that the infant is awake and performing certain activities or postures, for instance, playing with other children; participating in games with practice-learning about body parts (for example, “where is your hand?”); being carried in an adult’s arms; being attached to the caregiver’s body or in some carrying device (baby bag, sling, cradleboard, etc.); in a seating device (high chair, stroller, car seat, or any other type of seating device); in a walking device (walker, exersaucer, or any other type of device that provides help for the child to walk and/or support themself while standing up); in a playpen, or other similar equipment, bed or crib; in tummy time play.

The semi-structured interview used is available in Appendix A. All the data were anonymized and coded in order to protect the data.

### 2.4. Statistical Analysis

The analyses were carried out using the Statistical Package for the Social Sciences (SPSS) (IBM Corp, Armonk, NY, USA, version 22). Descriptive analyses among the four groups and for each group were performed. The data were expressed as the mean and standard deviation or the median and confidence interval. The categorical data were presented as absolute and relative frequencies. The data normality distribution was verified using Shapiro–Wilk’s Test.

In order to compare the groups, a multivariate analysis of variance (MANOVA) was conducted on all the variables which compose the dependent variables, an environmental factor, and the parental practices among the four groups (H_IT, R_IT, H_BR, and L_BR), with age as a covariate. After this step, post hoc comparisons were carried out to explore the pair group differences. Significance values, set up for *p*-value ≤ 0.05, were adjusted by the Bonferroni test. Effect sizes were calculated to avoid type II errors, the interpretation of which was based on eta-squared (η^2^): η^2^ = 0.01 indicates a small effect; η^2^ = 0.06 indicates a medium effect; η^2^ = 0.14 indicates a large effect [40].

In addition, correlation tests were performed, whereby the researchers investigate negative or positive associations between the variety of stimulation versus the environmental and parental practice variables. The tests considered not only the variety of stimulation total scores but also the correlation between each question. Coefficients of determination (r^2^) were also presented as effect sizes for the correlation tests and interpreted as insignificant (<0.10), weak (0.10 to 0.39), moderate (0.40 to 0.69), strong (0.70 to 0.89), and very strong (0.90 to 1.00) (Schober, Boer, Schwarte, 2018). 

### 2.5. Ethical Clearance

The study was approved by the human research ethics committee of the Federal University of São Carlos (CAAE: 04097718.9.0000.5504), and by the Tuscany Region Pediatric Ethics Committee (NCT01990183), following the resolution nº 510/2016 of the National Health Council and Declaration of Helsinki. All the caregivers accepted participating in the study through the consent form and providing written consent at the beginning of the data collection.

## 3. Results

### 3.1. Participants

A total of 115 infants (4.46 mean age ± 0.62 months, age range 3.09–5.95) participated in this study. The infants were distributed among the groups: H_IT (n = 34; mean age = 4.44 (±0.71)); R_IT (n = 23; mean age = 4.58 (±0.66)); H_BR (n = 31; mean age = 4.49 (±0.66)), and L_BR (n = 27 mean age = 4.34 (±0.63)) (Table 1).

### 3.2. Environment and Parental Practices

Regarding the environment and parental practices, the mother’s age (*p*-value < 0.01; ηp^2^ = 0.40; large effect size), education level (*p*-value < 0.01; ηp^2^ = 0.60; large effect size) and the education level of the father (*p*-value < 0.01; ηp^2^ = 0.49; large effect size), civil status (*p*-value = 0.01; ηp^2^ = 0.10; medium effect size), and variety of stimulation (*p*-value < 0.01; ηp^2^ = 0.27; large effect size) showed significant differences between the infants living in Brazil and those in Italy (Table 2), especially when there was an environmental risk factor, such as in the L_BR group, which obtained the lowest scores.

The post hoc analyses revealed that the mother’s age was statistically significantly different between the pair groups, except between the two healthy groups (H_IT and H-BR), and the variety of stimulation for motor development from AHEMD-IS was statistically significant between the pair groups, except between the two Italian groups (H_IT and R_IT). Moreover, the number of children and adults at home showed a significant difference between the groups, with the highest scores for the L_BR group (Table 3).

### 3.3. Environmental Factors and Parental Practices versus Variety of Stimulation

Regarding the total score of the variety of stimulation, positive correlations were found for (i) the maternal age (rho = 0.34, r^2^ = 0.11, *p* < 0.00); (ii) SES (rho = 0.48, r^2^ = 0.23, *p* < 0.00); and (iii) maternal and paternal education (rho = 0.42, r^2^ = 0.17, *p* < 0.00; rho = 0.28, r^2^ = 0.07, *p* < 0.00, respectively), while negative correlations were found only with the number of children at home (rho = −0.23, r^2^ = 0.05, *p* < 0.00). The variety of stimulation questions request information about the infants’ and children’s daily interactions. The variety of stimulation includes games that encourage the infant to learn about the body parts, the time that the infant spends carried in an adult’s arms, seated in a sitting device, standing in a standing device, e.g., walker, exersaucer, or any other type of device that provides help for the child to walk and/or support themself while standing up, in a playpen, playing in the tummy time position, and free to move, e.g., on the floor. Then, regarding these variables, the correlation analyses found positive correlations in the tummy time position, and the time that the infant is placed to move freely, with maternal age, and maternal and paternal education (Table 4).

## 4. Discussion

The present study verified that there is a significant difference in the parental practices and environments in healthy and at-risk infants at 3–6 months of age, living in upper-middle- (Brazil) and high-income (Italy) countries. The variety of stimulation for motor development was also different between infants living in Italy and Brazil, both in health and risk conditions. In high-income countries, the biological risk factor seems not to impact the variety of stimulation while, in the upper-middle-income countries, it is impacted by the environmental risk factors. It is crucial that, in high-income countries where the infants are well stimulated, they can continue to be stimulated also in risk conditions while, in the upper-middle-income countries, where the infants are already different for lower stimulation, the gap is even higher than when a risk situation is present. To our knowledge, so far there are no published studies that have evaluated these aspects, and this is a very crucial and novel finding of the present study.

Infants classified as having a low socioeconomic status, living in an upper-middle-income country (Brazil), presented low scores in maternal age, maternal and paternal education, and variety of stimulation, which may be explained by different access to quality education, scarce economic resources to survive, and consequently less knowledge about the stimulation of motor development [11,13].

The level of maternal and paternal education is an essential factor, not only for early childhood but also for school-age children, since parents with a high level of education adopt simple and easy parental practices, e.g., a disciplined routine of studies, which stimulates the potential for development and learning [41].

It is observed that there is gender inequality in the care of infants. Mothers are the protagonists of care more than fathers [42]. Consequently, it was observed that maternal education has more impact on child development when compared to paternal education [43].

In this context, in countries where there are education issues, i.e., UMIC, previous family knowledge on how to work horizontally with health professionals may be useful as orientation, and effective learning strategies (workshops, illustrative materials, practices on dolls, active methodology, and problem-based approaches) should be used to teach about a child’s motor skills development in order to improve and stimulate parental knowledge [44].

The World Health Organization and United Nations International Children’s Emergency Foundation encourage the implementation of responsive care and early learning to enhance early childhood development [8] and promote opportunities for full developmental potential, regardless of the risk exposure [34]. UNICEF has created some programs in UMICs in order to promote child development and empower caregivers to practice parenting positively, learning, responsive care, and stimulating the infant’s motor and cognitive abilities through websites and videos [45]. In addition, there are social and health programs especially for supporting growth and feeding [46]. However, in a big and diverse country like Brazil, the population may have not enough information, internet availability, and education [34] to access these programs. For instance, one of every four people in Brazil has no access to the internet. In total terms, this represents about 46 million people who are not able to access online health materials [47]. In this context, the local government needs to embrace the idea and provide the structure to allow the population to take advantage of these resources.

In addition, when it comes to planning public policies for early childhood, the managers should pay attention not only to mortality rates but also to stimulating the potential for child development, which is important for society to have an active and participating population. Overcoming mortality in the first year of life does not guarantee life and functionality, where wealth vs. vulnerability and opportunities vs. unemployment coexist, as in Brazil [48].

The present study observed that low maternal and paternal education is associated with poor parental practices and routines, such as the lack of games that encourage the infant to learn about parts of the body, the practice of the tummy time position, and if the parents leave the infant free to move, e.g., on the floor on a mattress.

The practice of the tummy time position showed important benefits for preventing obesity, a sedentary lifestyle, and plagiocephaly while improving motor performance, and the ability to move in prone, supine, crawling, and rolling positions [49]. However, parents must not only supervise but also interact and play with the infant, so that this is a pleasurable activity [50].

In Brazil, differently from Italy, it was observed that the caregivers never encourage infants in the tummy time position, although this activity is a component of the National Movement Guidelines in Australia, the United Kingdom, Canada, and South Africa [51,52,53,54], in the National Academy of Medicine, the American Academy of Pediatrics and the World Health Organization global guidelines for physical activity, sedentary behavior, and sleep for children <5 years of age. It is highly recommended that infants practice the tummy time position for at least thirty minutes, spread out over the course of every day [55].

Parental practices that encourage child development, such as face-to-face interaction, and leaving the infant free to move, are beneficial for the infant’s development and autonomy [56]. Another factor that is different between the analyzed countries is the maternal age groups. These data provide evidence that the mother’s age is a predictor of the risks, the older age for biological risk, and the younger age for environmental and parental practices risk.

The age difference between countries can be explained by differences in access to suitable quality education, gender inequality, lack of use or access to contraceptive methods, and participation in the labor market [57,58]. Even though the older mother has more risks of preterm birth, the maternal age showed a positive association with parental practices, e.g., the variety of stimulation, games that encourage the infant to learn about body parts, the practice of the tummy time position, space to move, learning time, and proposing suitable and recycled toys [59].

The present study shows some limitations. Firstly, even though the sample size is quite large, there is some heterogeneity in the different groups, as the different risk factors are environmental for the Brazilian groups, and biological for the Italian ones. Secondly, the semi-structured interview created by the authors was not validated, and other crucial variables were not investigated. In further studies, it might be interesting to consider other variables, such as the different types of parenting practices, the influence of kindergarten or home care, and to follow up by evaluating the motor and cognitive development of infants. It would also be interesting to consider how to implement the recommended parental practices, especially in UMIC.

## 5. Conclusions

Our results suggest that there are significant differences in the parental practices and environmental factors among infants living in low/middle- and high-income countries. Moreover, we have shown that risk factors affect parental practice only in the UMIC.

Low/middle-income countries presented low scores in the maternal age, maternal and paternal education, and variety of stimulation. The present study observed that low maternal and paternal education is associated with poor parental practices and routines such as, for instance, games that encourage the infant to learn about parts of the body, the practice of the tummy time position, and freedom for the infant to move, e.g., on the floor on a mattress. In low/middle-income countries, differently from in high-income countries, it was observed that the caregivers never encourage infants to practice the tummy time position. Since the home environment is the main stimuli for infant growth and development, our results are important for providing knowledge about the different cultures.

Overall, these findings highlight the importance of addressing new policies to improve awareness around the importance of providing enriched environments at home, especially in at-risk conditions, and also by means of technological tools [60,61].

## Figures and Tables

**Table 1 ijerph-19-10833-t001:** Characteristics of infants.

Variables	H_IT (n = 34)	R_IT (n = 23)	H_BR (n = 31)	L_BR (n = 27)
Mother of age mean (SD)	32.47 (3.47)	37.39 (5.35)	31.87 (7.10)	24.30 (5.67)
Number of children at home median (min–max)	1.00 (1.00–4.00)	2.00 (1.00–3.00)	2.00 (1.00–3.00)	2.00 (1.00–6.00)
Number of adults at home median (min–max)	2.00 (1.00–2.00)	2.00 (2.00–3.00)	2.00 (1.00–6.00)	2.00 (1.00–7.00)
Civil status/married people n (%)	15 (44.11%)	5 (21.73%)	28 (90.32%)	9 (29.03%)
Maternal education mean (SD)	5.41 (1.78)	4.74 (1.66)	4.65 (1.31)	0.89 (0.80)
Paternal education mean (SD)	4.85 (1.52)	4.78 (1.62)	4.26 (1.44)	1.41 (1.08)
Variety of stimulation score mean (SD)	12.79 (2.27)	13.30 (1.99)	11.68 (2.15)	9.74 (2.31)

Legend: healthy Italian infants (H_IT); Italian infants exposed to biological risk factors (R_IT); healthy Brazilian infants (H_BR); Brazilian infants exposed to environmental risk factors (L_BR).

**Table 2 ijerph-19-10833-t002:** Comparison between the groups.

	Error df	F	*p*-Value	Partial Eta Squared
Gestational age (weeks)	111.00	2.51	0.06	0.06
Age at assessment	111.00	0.65	0.58	0.02
M = 1; F = 2	111.00	0.57	0.63	0.02
Maternal age	111.00	24.50	0.00	0.40
Number of children at home	111.00	8.12	0.00	0.18
Number of the adults in the house	111.00	3.81	0.01	0.09
Maternal education: ISCED classification	111.00	56.44	0.00	0.60
Paternal education: ISCED classification	111.00	35.99	0.00	0.49
Civil status (1: married or civil union, 0: no)	111.00	4.15	0.01	0.10
SES (not Low: 1, low: 0)	111.00	57.42	0.00	1.00
Total score of Variety of Stimulation	111.00	13.84	0.00	0.27
Interpretation of Variety of Stimulation	111.00	11.20	0.00	0.23

Legend: (M) male; (F) female; (ISCED) International Standard Classification of Education; (SES) socioeconomic status.

**Table 3 ijerph-19-10833-t003:** Post hoc analyses in order to compare the group.

	L_BR vs. R_IT		L_BR vs. H_BR		L_BR vs. H_IT		H_BR vs. R_IT		H_BR vs. H_IT		H_IT vs. R_IT	
	Mean Difference [95% CI]	*p*-Value	Mean Difference [95% CI]	*p*-Value	Mean Difference [95% CI]	*p*-Value	Mean Difference [95% CI]	*p*-Value	Mean Difference [95% CI]	*p*-Value	Mean Difference [95% CI]	*p*-Value
Mother age	−13.10 [−17.3/−8.89]	0.00	−7.57 [−11.47/−3.68]	0.00	−8.17 [−11.99/−4.36]	0.00	−5.52 [−9.60/−1.45]	0.00	−0.60 [−4.28/−3.08]	1.00	−4.92 [−8.92/−0.92]	0.01
Number of children at home	1.06 [0.36/1.75]	0.00	0.86 [0.22/1.50]	0.00	1.02 [0.39/1.65]	0.00	0.20 [−0.48/0.87]	1.00	0.16 [−0.45/0.77]	1.00	0.04 [−0.62/0.70]	1.00
Number of adults in the house	0.47 [−0.14/1.07]	0.24	0.30 [−0.26/0.86]	0.94	0.67 [0.12/1.22]	0.01	0.17 [−0.42/0.76]	1.00	0.38 [−0.15/0.91]	0.36	−0.20 [0.78/0.37]	1.00
Maternal education: ISCED classification	−3.85 [−4.95/−2.75]	0.00	−3.76 [−4.78/−2.73]	0.00	−4.52 [−5.52/−3.52]	0.00	−0.09 [−1.16/0.98]	1.00	−0.77 [−1.73/0.20]	0.21	0.67 [−0.38/1.72]	0.53
Paternal education: ISCED classification	−3.38 [−4.46/−2.29]	0.00	−2.85 [−3.86/−1.84]	0.00	−3.45 [−4.44/−2.46]	0.00	−0.52 [−1.58/0.53]	1.00	−0.59 [−1.55/0.36]	0.58	0.07 [−0.97/1.11]	1.00
Civil status (1: yes, 0: no)	0.03 [−0.28/0.34]	1.00	−0.12 [−0.41/0.16]	1.00	0.23 [−0.05/0.51]	0.19	0.15 [−0.15/0.45]	1.00	0.35 [0.08/0.62]	0.01	−0.19 [−0.49/0.10]	0.46
SES (not low: 1, low: 0)	−1.00 [−1.00/−1.00]	0.00	−1.00 [−1.00/−1.00]	0.00	−1.00 [−1.00/−1.00]	0.00	0.00 [0.00/0.00]	1.00	0.00 [0.00/0.00]	1.00	0.00 [0.00/0.00]	1.00
Total score of Variety of Stimulation	−3.56 [−5.24/−1.89]	0.00	−1.94 [−3.49/−0.39]	0.01	−3.05 [−4.47/−1.53]	0.00	−1.63 [−3.25/0.00]	0.05	−1.12 [−2.58/0.35]	0.26	−0.51 [−2.10/1.08]	1.00
Interpretation of Variety of Stimulation	−1.35 [−2.08/−0.63]	0.00	−0.71 [−1.38/−0.03]	0.04	−1.22 [−1.88/−0.56]	0.00	−0.65 [−1.35/0.06]	0.09	−0.52 [−1.15/0.12]	0.19	−0.13 [−0.82/0.56]	1.00

Legend: healthy Italian infants (H_IT); Italian infants exposed to biological risk factors (R_IT); healthy Brazilian infants (H_BR); Brazilian infants exposed to environmental risk factors (L_BR); International Standard Classification of Education (ISCED); socioeconomic status (SES).

**Table 4 ijerph-19-10833-t004:** Pearson correlation between environmental factors and parental practices versus questions which compose the variety of stimulation dimension from AHEMD-IS.

	I/We Regularly (at Least Twice a Week) Play Games with my/our Child to Practice Learning about Body Parts. (e.g., Where Is Your Hand?)	My/our Baby Plays Regularly (at Least Twice a Week) with Other Children	Carried in Adult Arms, Attached to Caregiver’s Body or in some Carrying Device (Baby Bag, Sling, Cradleboard, etc).	In a Seating Device (High Chair, Stroller, Car Seat, or any Other Type of Seating Device).	In a Walking Device (Walker, Exersaucer, or any Other Type of Device that Provides Help for the Child to Walk and/or Support While Standing up).	In a Playpen, or Other Similar Equipment, Bed or crib.	In the Tummy Time Play.	Free to Move in any Space of the House	Total Score of Variety of Stimulation
Age at assessment	0.016	0.004	0.045	0.069	0.043	−0.001	0.041	0.270 **	0.203 *
Sex	0.019	−0.127	0.101	−0.171	−0.009	−0.074	−0.009	−0.008	−0.088
maternal age	0.137	0.093	0.095	−0.059	−0.015	−0.076	0.317 **	0.469 **	0.344 **
number of children at home	−0.288 **	0.211 *	0.049	−0.114	0.044	−0.129	−0.188 *	−0.190 *	−0.232 **
number of the adults in the house	0.019	0.019	−0.058	0.137	0.082	0.114	−0.176 *	−0.225 **	−0.071
Maternal schoolarity: ISCED classification	0.271 **	0.048	0.054	0.013	0.054	0.056	0.333 **	0.455 **	0.428 **
Paternal schoolarity:ISCED classification	0.205 *	0.086	0.006	0.017	0.063	0.020	0.199 *	0.363 **	0.287 **
civil status	−0.141	0.018	0.192 *	−0.078	−0.024	0.085	−0.096	−0.002	−0.019
SES	0.299 **	0.033	0.024	−0.004	0.048	0.003	0.420 **	0.531 **	0.484 **

Legend: healthy Italian infants (H_IT); Italian infants exposed to biological risk factors (R_IT); healthy Brazilian infants (H_BR); Brazilian infants exposed to environmental risk factors (L_BR). * Correlation is significant at the 0.05 level. ** Correlation is significant at the 0.01 level.

## Data Availability

The raw data supporting the conclusions of this article will be made available by the authors, without undue reservation.

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
