# Peer review of "Parental Practices and Environmental Differences among Infants Living in Upper-Middle and High-Income Countries: A Cross-Sectional Study"

_ijerph, 2022, doi:10.3390/ijerph191710833_

Round 1

Reviewer 1 Report

The purpose of the study was to evaluate parenting practices and environmental differences in healthy and at-risk infants (3-6 months of age) living in upper-middle-income (Brazil) and high-income (Italy) countries.

The abstract of the manuscript is functional. However, its structure could be improved. That is, it would be desirable for the method and results to be clearer and more specific.

The keywords are excessively broad (upper-middle-income countries, high-income countries). A revision and/or analysis according to the variables that guided the study is recommended.

The introduction and the theoretical-conceptual development of the manuscript is adequate and consistent. However, the background provided regarding the sociocultural context surrounding parenting and households in Brazil compared to Italy is superficial. Similarly, some of the contributions offered from the field of neuroscience are presented in a stilted manner.

The presentation of the method is consistent. Its rationale is pertinent. As a suggestion, 2 observations are made: a) Regarding the sample, it would be important to consider how the participants were finally selected and what exclusion criteria were considered; b) Regarding the data collection techniques, it would be important to specify how the semi-structured interview answered by the parents was constructed (areas of analysis, validation process, application procedure, etc.).

The data analysis strategy is relevant.

The ethical considerations that underpinned the study are specified in the work.

The presentation of results is adequate. The information is organized coherently according to the variables analyzed. It is recommended to verify that the tables are constructed according to the editorial standards of the journal.

Regarding the discussion, it is argued that the study identified a significant difference in the practices and environment surrounding healthy and at-risk kisses. They posit that motor stimulation presents differences between babies residing in Brazil and Italy, considering their health conditions, socioeconomic level, mother's age, at-risk conditions, etc. In addition, a greater bonding in the upbringing of the babies by the mothers is observed. In fact, it was observed that low maternal and paternal education is related to poor parenting practices and routines. Despite the interesting reflections raised in this section, it is striking that the depth of the arguments is weak. There is ample scientific evidence that shows conclusively that the conditions of parenting, support and parental supervision observed in Latin American countries are different from those observed in European Union countries, considering the existing differences in terms of parents' educational level, socioeconomic level, availability of support networks, strengthening of parental competencies, quality of life, access to health, etc. It is also evident that public policies aimed at promoting parental care, positive parenting, co-parenting or breastfeeding care are still precarious in several Latin American countries (there are multiple reports written by UNESCO, UNICEF, WHO, World Bank, etc.). Overall, it would be desirable that the discussion reaches a greater depth in its arguments, seeking a critical analysis of the most significant results provided by the study.

The limitations of the study are presented in a superficial manner. It requires structural improvement, taking into account the shortcomings potentially identified in the method.

The conclusions are weak. They do not clarify, in depth, whether the results obtained manage to comprehensively respond to the purpose of the study.

Author Response

Dear reviewer,
we would like to thank you for your attention to revising the article. You pointed out important topics in all sections, it is essential for us. We appreciated your comments, and every change in the paper was highlighted in yellow, the final file was attached to the journal’s platform. 
Thank you, 
Best regards, 
All authors.

Reviewer 2 Report

Good work! Congratulations!

1. The main question is the study of parental practices and environment differences in infants in different social and biological conditions in 2 countries to analyse their possible impact in the child neurodevelopment and being able to plan social interventions. 2. The topic is really relevant due to the high influence of costumes and care practices in neurodevelopment. 3. it adds the cross correlations among high or low income countries and biological risk factors as prematurity. 4. I have no specific improvements to propose in methodology. Maybe in the design of next studies it could include the influence of kinder-garden or home care in the variables, or it would be good to assess motor or global development in the same children, in a follow up. 5. In my opinion conclusions are consistent with initial questions. It is not a complex study but the conclusions are well honest and clear to the purpose. 6. Maybe other references could be added on the discussion, which is the weakest part, however it is good enough in my opinion.
7. No comments in tables or figures

Author Response

(The authors gave the same response as above.)
